

# The impact of solar radiation on polar mesospheric ice particle formation

Mario Nachbar[1], Henrike Wilms[2], Denis Duft[1], Tasha Aylett[3], Kensei Kitajima[4], Takuya Majima[4], John M. C. Plane[3], Markus Rapp[2,5], and Thomas Leisner[1,6]

[1]Institute of Meteorology and Climate Research, Karlsruhe Institute of Technology – KIT, Hermann-von-Helmholtz-Platz 1, 76344 Eggenstein-Leopoldshafen, Germany

[2]Deutsches Zentrum für Luft- und Raumfahrt, Institut für Physik der Atmosphäre, Oberpfaffenhofen, Germany

[3]School of Chemistry, University of Leeds, Leeds, UK, LS2 9JT

[4]Department of Nuclear Engineering, Kyoto University, Kyoto 615-8540, Japan

[5]Meteorologisches Institut München, Ludwig-Maximilians-Universität München, Munich, Germany

[6]Institute of Environmental Physics, University of Heidelberg, Im Neuenheimer Feld 229, 69120 Heidelberg, Germany

*Correspondence to*: Mario Nachbar (Mario.nachbar@kit.edu)


**Abstract**

Mean temperatures in the polar summer mesopause can drop to 130 K. The cold temperatures in combination with water vapor mixing ratios of a few parts per million give rise to the formation of ice particles. These ice particles may be observed as polar mesospheric clouds. Mesospheric ice cloud formation is believed to initiate heterogeneously on small aerosol

particles ($r < 2$ nm) composed of re-condensed meteoric material, so called Meteoric Smoke Particles (MSPs). Recently, we investigated the ice activation and growth behavior of MSP analogues under realistic mesopause conditions. Based on these measurements we presented a new activation model which largely reduced the uncertainties in describing ice particle formation. However, this activation model neglected the possibility that MSPs heat up in the low density mesopause due to absorption of solar and terrestrial irradiation. Radiative heating of the particles may severely reduce their ice formation

ability. In this study we expose MSP analogues ($Fe_2O_3$ and $Fe_xSi_{1-x}O_3$) to realistic mesopause temperatures and water vapor concentrations and investigate particle warming under the influence of variable intensities of visible light (405, 488, and 660 nm). We show that Mie theory calculations using refractive indices of bulk material from the literature combined with an equilibrium temperature model presented in this work predict the particle warming very well. Additionally, we confirm that the absorption efficiency increases with the iron content of the MSP material. We apply our findings to mesopause

conditions and conclude that the impact of solar and terrestrial radiation on ice particle formation is significantly lower than previously assumed.




## 1 Introduction

The coldest temperatures in the terrestrial atmosphere are encountered in the polar summer mesopause, where mean daily temperatures at high latitudes can fall to below 130 K (Lübken, 1999; Lübken et al., 2009). These low temperatures in combination with $H_2O$ concentrations of a few parts per million (Hervig et al., 2009; Seele and Hartogh, 1999) lead to highly

supersaturated conditions which allow for the formation of ice particles (e.g. Lübken et al., 2009; Rapp and Thomas, 2006). When the ice particle radii reach about 30 nm and their concentration is of the order of $100\ cm^{-3}$ they become optically visible and may be observed as Polar Mesospheric Clouds (PMCs) (e.g. Rapp and Thomas, 2006). When observed from ground, these clouds are often referred to as Noctilucent Clouds (NLCs). Because of their particular wavy appearance and their high elevation of about 83 km, PMCs have received much attention since their first reported observation in 1885

(Leslie, 1885). The current scientific interest in these extraordinary clouds is substantiated in their potential role as tracer for the dynamical structure of the summer mesopause (e.g. Demissie et al., 2014; Kaifler et al., 2013; Rong et al., 2015; Witt, 1962) or for long term trends caused by anthropogenic emissions of $CO_2$ and $CH_4$ (e.g. Hervig et al., 2016; Thomas and Olivero, 2001; Thomas et al., 1989). However, in order to use observation of PMCs as a tracer, an in-depth understanding of the processes involved in PMC formation is necessary.

Wilms et al. (2016) found that next to dynamical processes the description of the initial formation of the ice particles significantly affects modeled PMC properties. Ice particle formation is believed to initiate heterogeneously on nanometer sized re-condensed meteoric material, so called Meteoric Smoke Particles (MSPs) (e.g. Gumbel and Megner, 2009; Keesee, 1989; Rapp and Thomas, 2006; Turco et al., 1982). This conjecture is strongly supported by satellite and rocket born observations showing that MSPs are included in PMC ice particles (Antonsen et al., 2017; Havnes et al., 2014; Hervig et al.,

2012). The initial ice particle formation has been described in two different ways. Either activation-barrier free growth is assumed to set in for saturations in excess of the equilibrium saturation over the curved particle surface (e.g. Berger and Lübken, 2015; Schmidt et al., 2018), or ice particle formation is described using classical nucleation theory (e.g. Asmus et al., 2014; Bardeen et al., 2010; Rapp and Thomas, 2006; Wilms et al., 2016). Both approaches assume the formation of hexagonal ice and the latter typically requires much higher critical saturations to initiate ice particle growth. In order to

reduce the large uncertainties in describing the initial formation of PMC ice particles we designed a laboratory experiment to study ice particle formation under realistic mesopause conditions (Duft et al., 2015). Recently, we investigated the ice activation and growth behavior on $SiO_2$, $Fe_2O_3$ and mixed iron silicate nano-particles which serve as analogues for MSPs. We found that the primary ice phase forming on the MSP analogues at the conditions of the summer mesopause is Amorphous Solid Water (ASW) (Nachbar et al., 2018b; Nachbar et al., 2018c). Additionally, we showed that MSPs adsorb

up to several layers of water until ice growth activates as soon as the saturation exceeds the saturation vapor pressure of ASW including the Kelvin effect for the ice-covered or "wet" particle radius, and considering the collision radius of water molecules (Duft et al., 2018).

Asmus et al. (2014) pointed out that MSPs may heat up in the low density atmosphere of the mesopause by absorbing solar and terrestrial irradiation. The extent of this effect depends on the MSP composition and has been proposed to increase with

increasing iron content. Interestingly, satellite and rocket-born investigations indicate that MSPs are most likely composed of iron rich materials such as magnetite ($Fe_3O_4$), wüstite (FeO), magnesiowüstite ($Mg_xFe_{1-x}O$, x=0-0.6), and iron-rich olivine ($Mg_{2x}Fe_{2-2x}SiO_4$, x=0.4-0.5) (Hervig et al., 2017; Rapp et al., 2012). Using nucleation theory, Asmus et al. (2014) concluded that the warming for such MSP materials significantly impacts the ice-forming ability of the particles, thus rendering them ineffective nuclei. However, up until now this conclusion has not been confirmed experimentally. To this end, we extended

our experimental setup with a laser system which allows MSP analogues to be exposed to a known intensity of visible light at three wavelengths (405, 488, and 660 nm). We studied the number of adsorbed $H_2O$ molecules on $Fe_2O_3$ and $Fe_xSi_{1-x}O_3$ nanoparticles under the influence of the laser light at controlled gas-phase $H_2O$ concentration and background pressure. In this way, we could determine the offset of the particle temperature from the ambient temperature. The experimental method





is described in more detail in Sect. 2. From the experimentally determined temperature offsets we then deduce absorption efficiencies using a light absorption model, which is introduced in Sect. 3. In Sect. 4, we present our results on the absorption efficiencies and compare them to Mie theory calculations using literature values of the refractive indices. We estimate the maximum temperature offset for MSPs in the summer mesopause and discuss the consequences on ice particle formation.

Finally, we summarize our main conclusions in Sect. 5.

## 2 Experimental Method

We produce singly charged $Fe_2O_3$, $SiO_2$ or $Fe_xSi_{1-x}O_3$ particles with radii smaller than 4 nm in a non-thermal low pressure microwave plasma particle source (Nachbar et al., 2018a). The particles are transferred online into the vacuum system of the experiment (illustrated in Fig. 1) which has been described in detail elsewhere (Duft et al., 2015; Meinen et al., 2010;

Nachbar et al., 2018b; Nachbar et al., 2016).

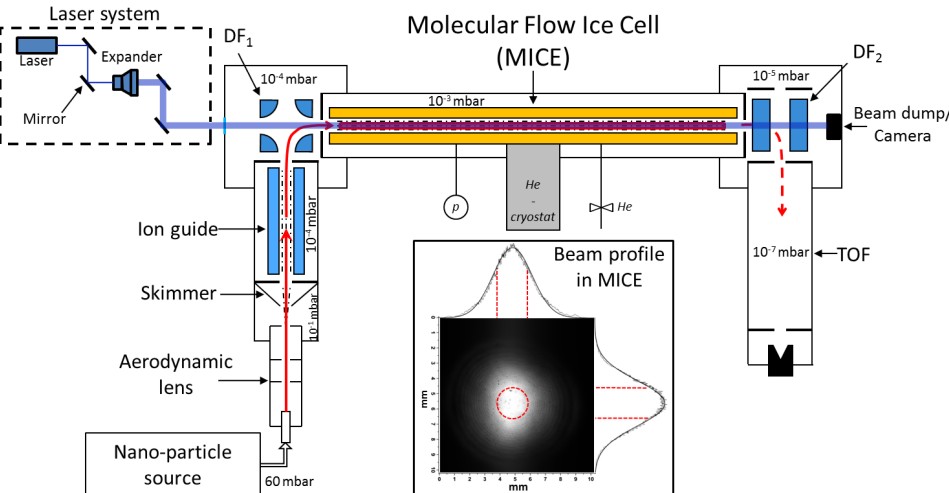

**Figure 1.** Illustration of the experimental setup. The insert shows a camera image of the laser beam profile (λ=488 nm) taken at the exit of MICE. Vertical and horizontal cross sections of the laser beam profile are shown above and right of the image, overlaid with fitted Gaussian curves. The red dashed lines indicate the radial extent of the levitated nano-particle cloud. See text for more
details.

In brief, singly charged nanometer-sized particles enter the vacuum chamber through an aerodynamic lens and a skimmer. After the skimmer, the particles enter an rf-octupole serving as an ion guide. The particles are mass selected ($\Delta m/m \leq 7\%$) with an electrostatic quadrupole deflector ($DF_1$) and subsequently enter into the Molecular flow Ice Cell (MICE). MICE is a modified quadrupole ion trap. A temperature-controlled He environment at a pressure of $(1-5)\cdot 10^{-3}$ mbar thermalizes

the particles under molecular flow conditions. The helium pressure is adjusted with a leak valve attached to a helium cylinder (99.999% purity) and the pressure is measured and corrected (Yasumoto, 1980) using a pressure sensor (Ionivac ITR 90). In MICE, the particles also interact with a well-defined concentration of gas-phase $H_2O$ molecules (Duft et al., 2015). For a typical experiment, MICE is filled with $10^7$ particles in about 1 s, followed by storing of the particles for up to several hours. Depending on the conditions applied in MICE, $H_2O$ molecules adsorb on the particles until an equilibrium

state is reached, or ice growth initiates on the particles. These processes are monitored by periodically extracting a small portion of the trapped particle population from MICE. After extraction, the particles are accelerated orthogonally into a time-of-flight spectrometer for mass measurement (TOF).

The setup has been extended with a laser system equipped with three lasers of different wavelengths λ=405 nm (Obis LX 405), λ=488 nm (Obis LX 488), and λ=660 nm (Obis LX 660). A combination of a laser beam expander (Edmund Optics




10X VIS broadband beam expander), mirrors, and a quartz glass window guides the expanded laser beam horizontally through the center of MICE pointing onto a beam dump. The light intensity in MICE was calibrated by measuring the power and the beam profile in MICE with a power-meter (Coherent PM USB PS19Q) and a CCD camera (Thorlabs 4070M-GE-TE). A typical beam profile of the expanded 488 nm laser beam is shown in the insert of Fig. 1. The red dashed lines indicate

the maximum ion cloud diameter $d = 2$ mm calculated for the combinations of particle mass ($2 \cdot 10^4$ Da $- 50 \cdot 10^4$ Da), RF frequency (30 kHz – 1000 kHz), and amplitude (200 V – 1000 V) applied in the present study (Majima et al., 2012). Allowing for a misalignment of the laser beam of up to 0.5 mm from the ion trap center, we conclude that the particles are located within a diameter of 3 mm from the center of the expanded laser beam. We use the mean of the intensity values at $d = 3$ mm and the center of the laser beam to describe the light intensity irradiating the particles. The uncertainty is defined

by the difference between the intensity value at the center of the laser beam and the mean value.

In this work, we apply conditions with saturations below the threshold for ice growth, i.e. where only adsorption occurs. Each experiment begins by filling MICE with a fresh charge of nano-particles. The initially dry nano-particles adsorb $H_2O$ molecules until an equilibrium state between adsorbing $H_2O$ flux and desorbing flux is reached. The process of reaching the equilibrium state is illustrated in Fig. 2a which shows the time-evolution of the particle mass for $Fe_2O_3$ particles with a dry

particle radius $r_{dry} = \left(3m_0/4\pi\rho_p\right)^{1/3} = 2.9$ nm ($\rho_p = 5.2$ gcm$^{-3}$), a $H_2O$ gas phase concentration $n_{H2O} = 1.1E16$ m$^{-3}$, and a temperature of the environment surrounding the particles $T_{env} = 148.8$ K.

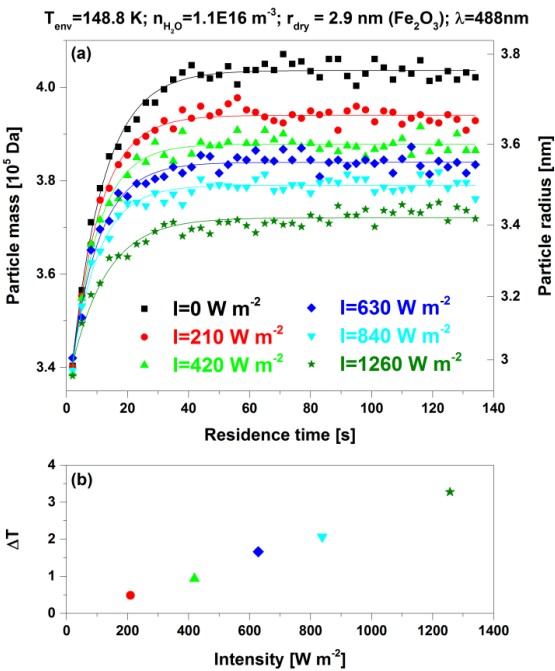

Figure 2. Panel a: Mean particle mass as a function of residence time in MICE for 6 different mean light intensities (λ=488 nm, $r_{dry}$ =2.9 nm ($Fe_2O_3$), $T_{env}$=148.8 K, $n_{H2O}$=1.1E16 m$^{-3}$). The solid curves represent fits of Eq. (2) to the data. Panel b: Particle
temperature offsets $\varDelta T$ (Eq. (4)) as a function of the mean light intensities.

The black squares show the adsorption curve without light irradiation for which $T_{env}$ equals the particle temperature $T_p$. If the particles are heated by light irradiation ($T_p > T_{env}$), the water molecule flux desorbing from a particle increases which causes a reduction in the number of adsorbed $H_2O$ molecules in equilibrium. This effect is shown by the coloured data for illumination with the 488 nm laser at various mean light intensities. Figure 2b shows the corresponding particle temperature

offsets $\varDelta T$ which were determined by analyzing the steady-state mass of adsorbed water molecules. The method for deriving $\varDelta T$ is presented below.




In equilibrium, the adsorbing flux density $j_{ads}$ and desorbing flux density $j_{des}$ of water molecules are equal (Duft et al., 2018):

$$\underbrace{\frac{n_{H2O} \cdot v_{th}}{4}}_{j_{ads}} = \underbrace{c_{H2O} \cdot f \cdot \exp\left(-\frac{E_{des}^0}{RT_p} + \frac{2\sigma v}{RT_p r_{dry}}\right)}_{j_{des}} \qquad (1)$$

Here, $n_{H2O}$ is the number concentration of water molecules in the gas phase, $c_{H2O} = m_{ads}/\left(m_{H2O} \cdot A_p\right)$ is the concentration of adsorbed water molecules on the surface of the nanoparticles, $m_{H2O}$ is the mass of a water molecule, $A_p = 4\pi r_{dry}^2$ is the

surface area of the dry nano-particle, $f=10^{13}$ Hz is the vibrational frequency of a water molecule on the particle surface (Pruppacher and Klett, 2010), $v_{th} = \sqrt{8kT_{env}/\pi m_{H2O}}$ is the mean thermal velocity of gas phase $H_2O$ molecules, $R$ is the ideal gas constant, $v = 6.022 \cdot 10^{23} \cdot m_{H2O}/\rho_{ice}$ is the molar volume of a $H_2O$ molecule, and $E_{des}^0$ represents the mean desorption energy of a $H_2O$ molecule for a planar surface of the particle material. The density of ASW and crystalline ice are very similar at the temperatures under investigation (Brown et al., 1996; Loerting et al., 2011). Here, we use the

parameterization for crystalline ice $\rho_{ice}[g/cm^3] = 0.9167 - 1.75 \cdot 10^{-4} \cdot T_p[°C] - 5 \cdot 10^{-7} \cdot \left(T_p[°C]\right)^2$ (Pruppacher and Klett, 2010). For the surface tension of ASW we use $\sigma[mN\,m^{-1}] = (114.81 - 0.144 \cdot T[K])$ which is based on an extrapolation of experimental data for supercooled water (Nachbar et al., 2018c). The adsorbed mass of $H_2O$ molecules in equilibrium $m_{ads}$ and the initial particle mass $m_0$ are determined by an exponential fit (represented by the solid curves in Fig. 2a) of the following form:

$$m(t) = m_0 + m_{ads} \cdot \left(1 - exp\left(\frac{-t_{res}}{\tau}\right)\right) \qquad (2)$$

The radius of the wet particle is indicated by the right ordinate in Fig. 2a and follows from the measured particle mass according to:

$$r_{wet} = \left(r_{dry}^3 + \frac{3}{4\pi}\frac{m_{ads}}{\rho_{ice}}\right)^{1/3} \qquad (3)$$

The particle temperature offset $\Delta T$ is determined by a set of two measurement runs, one without and one with light

illumination. The mean desorption energy for a $H_2O$ molecule is determined from the measurement without light illumination by solving Eq. (1) for $E_{des}^0$. For the data shown in Fig. 2, this procedure results in $E_{des}^0 = 42.52$ kJ mol$^{-1}$. We only analyse data with coverages above 1 mono-layer for which the desorption energy is expected to depend only weakly on $H_2O$ coverage (Mazeina and Navrotsky, 2007; Navrotsky et al., 2008; Sneh et al., 1996). For such coverages we did not observe any significant influence of the $H_2O$ coverage or the particle temperature on the values of $E_{des}^0$ determined in our

previous study (Duft et al., 2018). Therefore, we can determine the particle temperature under light illumination assuming a constant $E_{des}^0$-value. Rearranging Eq. (1) yields:

$$T_p = T_{env} + \Delta T = \left(E_{des}^0 - \frac{2\sigma v}{r_{dry}}\right)\Big/\left(R \cdot ln\left(\frac{m_{ads}(T_p) \cdot f}{m_{H2O}\pi n_{H2O}v_{th}r_{dry}^2}\right)\right) \qquad (4)$$

Since $\sigma$ and $\rho_{ice}$ are dependent on the particle temperature, Eq. (4) should be solved numerically. However, a sensitivity analysis has shown that calculating $\Delta T$ analytically using constant $\sigma(T_{env})$ and $\rho_{ice}(T_{env})$ deviates less than 1 % from the

numerical solution. We therefore analysed our data using $\sigma(T_{env})$ and $\rho_{ice}(T_{env})$. The determined temperature offset $\Delta T$ can be used with an equilibrium temperature model to calculate the light absorption efficiency of the particles $Q_{abs}$. The equilibrium temperature model is introduced in the next section.

### 3 Equilibrium temperature model

The equilibrium temperature of particles levitated in MICE is described by a balance between power sources and sinks.

Sources are absorption of laser light $P_\lambda^a$ and of infrared radiation emitted by the environment $P_{env}^a$. Sinks are cooling due to collisions with the He background gas $P_{col}$ and black body radiation of the particle in the infrared $P_{rad}^e$. Note that in equilibrium, the heat from sublimation and condensation cancels. The balance equation is:





$$P_\lambda^a + P_{env}^a = P_{rad}^e + P_{col} \tag{5}$$

The absorption of laser light with intensity $I$ depends on the material and wavelength-dependent absorption efficiency of the particles $Q_{abs}$. The absorption efficiency is defined as the absorption cross section divided by the geometrical cross section $A_{geo} = \pi r_{dry}^2$. We assume the water ad-layer to be entirely transparent to visible light so that $P_\lambda^a$ is the power absorbed by

the MSP alone:

$$P_\lambda^a = I \cdot Q_{abs}(\lambda, r_{dry}) \cdot A_{geo} \tag{6}$$

For the cooling due to collisions with the He gas we use the description presented in Asmus et al. (2014):

$$P_{col} = A_{col} \cdot \frac{\alpha \cdot p}{4kT_{env}} v_{th} \cdot k \frac{\gamma+1}{2(\gamma-1)} \cdot \Delta T \tag{7}$$

For particles in the nanometer regime, the collision surface area $A_{col}$ must include the radius of the colliding He atom

($r_{He} = 0.14\,nm$, Bondi, 1964), so that $A_{col} = 4\pi(r_{wet} + r_{He})^2$. The helium pressure in MICE is represented by $p$, $v_{th} = \sqrt{8kT_{env}/\pi m_{He}}$ is the mean thermal velocity of He atoms, $\gamma$ is the heat capacity ratio, $\Delta T = T_p - T_{env}$ is the temperature difference between the particle and the environment, and $\alpha$ is the thermal accommodation coefficient. The particles investigated in this work are water covered metal oxides. The thermal accommodation coefficient of He on comparable surfaces has been measured to be $0.525 \pm 0.125$ (Fung and Tang, 1988; Ganta et al., 2011).

For the conditions applied in MICE, $P_{rad}^e$ and $P_{env}^a$ are several orders of magnitude smaller than $P_\lambda^a$ and $P_{col}$ and can be neglected in the analysis of the experimental results. For mesospheric conditions, these two terms may be calculated as presented in Asmus et al. (2014). Substituting Eq. (6) and Eq. (7) in Eq. (5) and solving for the absorption efficiency $Q_{abs}(\lambda, r_{dry})$ yields:

$$Q_{abs}(\lambda, r_{dry}) = \frac{A_{col}}{I \cdot A_{geo}} \alpha p v_{th} \frac{(\gamma+1)}{8T_{env}(\gamma-1)} \Delta T \tag{8}$$

## 4 Results and discussion

### 4.1 Absorption efficiencies

$H_2O$ adsorption measurements similar to those presented in Fig. 2 were recorded for $Fe_2O_3$ particles with dry particle radii between 1.3 and 3.2 nm. Particle temperature offsets $\Delta T$ were determined from the equilibrium adsorption measurements according to Eq. (4) and converted to absorption efficiencies for each light intensity according to Eq. (8). The absorption

efficiencies for each set of experiments with the same dry particle radius were averaged. The results are shown in Fig. 3 as a function of dry particle radius on a double logarithmic scale. The main measurement uncertainties originate from the inhomogeneous intensity profile of the expanded laser beam and the uncertainty of the thermal accommodation coefficient $\alpha$, which are systematic error sources. The error bars shown for the particle radius represent the width of the particle size distribution. In order to compare our data to $Q_{abs}$ values derived from literature data of the complex refractive index $m =$

$n + i \cdot k$, $Q_{abs}$ was calculated from the extinction efficiency $Q_{ext}$ and from the scattering efficiency $Q_{scat}$ using Mie theory (Bohren and Huffmann, 2007) according to:

$$Q_{abs}(\lambda, r_{dry}, m) = Q_{ext}(\lambda, r_{dry}, m) - Q_{scat}(\lambda, r_{dry}, m) \tag{9}$$

Note that the size parameter $x = 2\pi r_{dry}/\lambda$ is much smaller than 1 (Rayleigh regime) for all particle sizes investigated in the present study and that the absorption cross section for such size parameters is proportional to the volume of the particle and

thus $Q_{abs}$ is proportional to $r_{dry}$.

Large differences exist throughout the literature for the imaginary part of the refractive index for hematite ($Fe_2O_3$) in the visible and near-infrared (Zhang et al., 2015). These differences increase with increasing wavelength and reach a factor of 40 at $\lambda$=660 nm. In order to compare our results with literature data, we collated all works which determined the real part $n$ and the imaginary part $k$ of the refractive index for $Fe_2O_3$ between 400 and 700 nm (Bedidi and Cervelle, 1993; Hsu and




Matijević, 1985; Longtin et al., 1988; Querry, 1985). Note that the data reported in Longtin et al. (1988) are based on measurements from Kerker et al. (1979). We calculated the absorption efficiencies for all sets of refractive indices as a function of the particle radius. The results are shown by the dashed and dotted curves in Fig. 3 with the colours indicating the wavelength. The solid lines are mean values of the absorption efficiencies calculated from literature data. Note that the data

obtained with the refractive indices from Bedidi and Cervelle (1993) and Longtin et al. (1988) are identical at 488 nm. Our data at λ=405 nm agree well with the Mie theory calculations using the literature refractive indices, except for Bedidi and Cervelle (1993). Our data for 488 nm are also in good agreement with literature, except for the calculations using the refractive indices from Querry (1985). At 660 nm our experimental results lie within the large scatter of the literature values. The absorption efficiencies deduced from the refractive indices of Hsu and Matijević (1985) and Longtin et al. (1988) are

smaller than our results, whereas the values deduced from Querry (1985) and Bedidi and Cervelle (1993) are larger. The latter is supported by the work of Meland et al. (2011) who concluded from angle-resolved light scattering experiments with hematite particles that the $k$-values of Querry (1985) and Bedidi and Cervelle (1993) are too high.

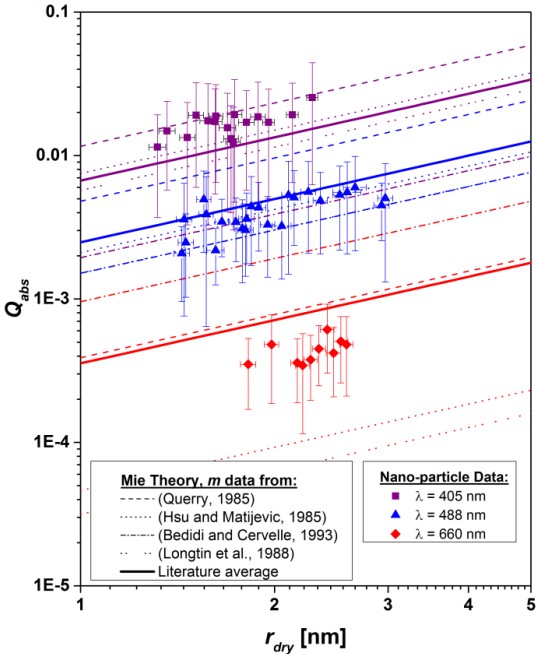

**Figure 3. Absorption efficiencies as a function of dry particle radius for Fe$_2$O$_3$ nano-particles at λ=405 nm, λ=488 nm, and**
**λ=660 nm. The dotted and dashed curves represent Mie theory calculations using refractive indices from literature with the colors indicating the wavelength.**

Overall, the experimentally determined absorption efficiencies show a linear trend with particle radius (compare to Mie theory calculations) and are within the spread of Mie theory calculations using the literature refractive indices. We therefore conclude that our method of determining $Q_{abs}$ from the equilibrium temperature of nano-particles via the amount of adsorbed

water is validated. Furthermore, we conclude that the equilibrium temperature model presented in this work can be used with literature values of bulk refractive indices of potential MSP materials to estimate the equilibrium temperature of MSPs in the mesopause.

Asmus et al. (2014) proposed that the temperature increase of MSPs due to absorption of solar irradiation increases linearly with increasing iron content of the particle material. In order to experimentally test this hypothesis, we measured the

absorption efficiency at λ=488 nm (the laser wavelength closest to the maximum of the solar irradiation) for Fe$_2$O$_3$ and iron-silicate particles Fe$_x$Si$_{1-x}$O$_3$ (0 < x < 1) ($r_{dry}$ = 2 nm) of varying iron content. The results are shown in Fig. 4 together with





Mie theory calculations using the refractive indices for $Mg_xFe_{1-x}SiO_3$ (Dorschner et al., 1995), FeO (Henning et al., 1995), FeOOH (Bedidi and Cervelle, 1993), and the mean value for $Fe_2O_3$ from Fig. 3. The data support the assumption that $Q_{abs}$ depends linearly on the iron content. Consequently, the potential MSP material which would heat up the most is FeO with a stoichiometric iron content of 0.5.

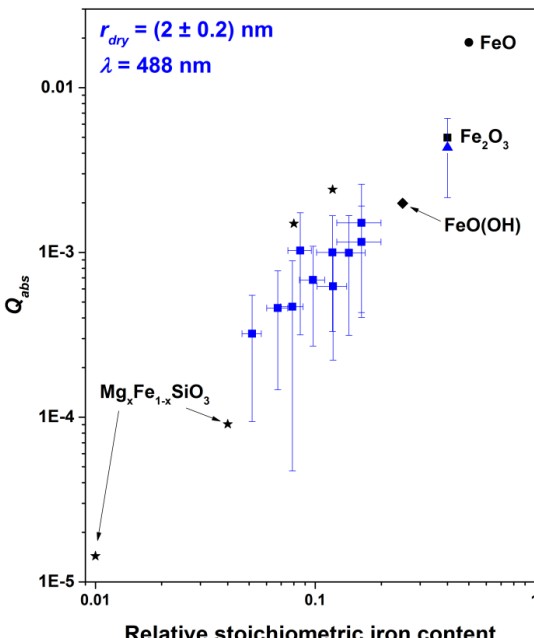

**Figure 4.** Absorption efficiencies for $r_{dry}$=2 nm particles at λ=488 nm of various particle materials as a function of the relative stoichiometric iron content. Blue squares and the triangle represent experimental results for $Fe_xSi_{1-x}O_3$ (0 < x < 1) and $Fe_2O_3$ particles, respectively. The black circle, square, diamond, and stars represent Mie theory calculations for FeO, $Fe_2O_3$, FeO(OH), and $Mg_xFe_{1-x}SiO_3$ particles using refractive indices from literature (see text).

**4.2 The impact of solar radiation on ice particle formation**

In this section we discuss the impact of solar radiation on the critical temperature of the environment $T_{cr,env}$ needed to activate ice growth. To this end we combine our previously presented ice growth activation model (Duft et al., 2018) with the equilibrium temperature model of this work. A description of the method can be found in Appendices A and B. In order to estimate the maximum impact of solar radiation, we assume that the particles are composed of FeO, the potential MSP material with the highest iron content and which is therefore expected to heat up the most. To calculate the water coverage on FeO particles we use an energy of desorption $E_{des}^0 = 42.7$ kJ/mol, which was previously determined for $Fe_2O_3$ particles (Duft et al., 2018). For the incoming solar irradiation we use the maximum solar zenith angle of 46.5°(21st June, noon) at 69°N, a typical latitude of MSP observations. In Fig. 5a we compare calculated critical temperatures as a function of the MSP radius with and without particle heating by solar irradiation for an altitude of 87 km (0.27 Pa). Here, we assume a constant $H_2O$ mixing ratio of 3 ppm which is typical for the polar summer mesopause (Hervig et al., 2009).

The solid blue curve in Fig. 5a shows results obtained when neglecting solar heating of the particles ($T_p=T_{env}$), i.e. representing non-absorbing particles. The horizontal dotted line at T=130 K indicates the measured mean temperatures at 87 km and 69°N during June and July (Lübken, 1999). In combination with typical temperature variations on the order of 10 K (Rapp et al., 2002) we conclude that the atmospheric temperature of the summer mesopause frequently falls below the critical temperature of non-absorbing MSPs with particle radii above 0.5 nm. For comparison, the dashed black curve shows calculated critical temperatures using the Kelvin effect of hexagonal ice at the dry particle radius, which represent the highest




activation temperatures currently assumed in mesospheric models (e.g. Berger and Lübken, 2015; Schmidt et al., 2018). For $r_{dry} < 1.1$ nm, the size of most MSPs in the polar summer mesopause (Bardeen et al., 2010; Megner et al., 2008a; Megner et al., 2008b; Plane et al., 2014), the new activation model excluding solar irradiation predicts ice particle formation at higher temperatures than currently assumed in models. This is explained by the uptake of water molecules by the MSPs which

increases the particle size and therefore causes a reduction of the Kelvin effect. This effect outweighs the vapor pressure difference between ASW and hexagonal ice for particle radii smaller than 1.1 nm.

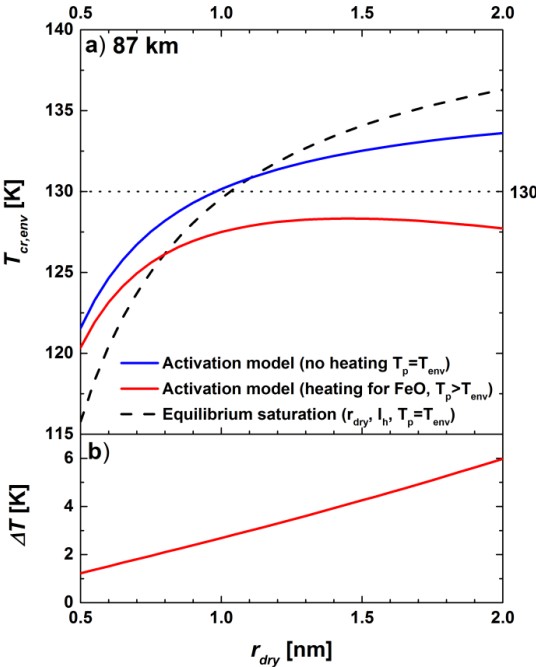

**Figure 5.** Critical temperature (panel a) and particle temperature offset (panel b) at 87 km altitude for FeO particles as a function of dry particle radius. The blue curve represents critical temperatures calculated with the activation model neglecting solar
irradiation and the red curves consider solar irradiation. For comparison, we show the equilibrium saturation calculated using the Kelvin equation for hexagonal ice with the dry particle radius by the dashed black curve.

The solid red curve in Fig. 5a shows results obtained when including solar heating of the particles ($T_p > T_{env}$) yielding lower $T_{cr,env}$ values. Figure 5b shows the offset of the particle temperature from the ambient temperature $\Delta T$ at critical conditions. These values are almost identical to the difference in critical temperatures between the activation model without and with

solar heating. We find that even the most absorbing particles with radii below 1.5 nm warm by less than 4 K at 87 km altitude. The particle heating will be much lower for other MSP materials and at lower altitudes due to the higher collisional cooling rate at higher pressures. In general, the particle heating reported here is about 5 times less than previous estimates (Asmus et al., 2014) for two main reasons: (1) the uptake of water molecules increases the particle surface area and therefore the collisional cooling rate; and (2) the thermal accommodation coefficient of $\alpha=0.5$ used in previous calculations (see also

Espy and Jutt (2002) and Grams and Fiocco (1977)) is very likely an underestimation. We use a value of 1 based on recent results of laboratory experiments which increases the collisional cooling rate by a factor of 2 (see Appendix B for more details).

## 5    Summary and Conclusions

We have presented $H_2O$ adsorption measurements on MSP analogues ($Fe_2O_3$ and $Fe_xSi_{1-x}O_3$ nano-particles) exposed to

variable intensities of visible light at 405, 488, and 660 nm. The experiments were performed at particle temperatures and





H$_2$O concentrations representative for the polar summer mesopause and the visible light covers the maximum in the solar irradiance. The reduction in the number of adsorbed water molecules under irradiation allows direct determination of the particle temperature increase caused by light absorption. We used the measured temperature increase in an equilibrium temperature model to determine the absorption efficiency of the particles. The results show that the equilibrium temperature

model is applicable and it can be used with literature values of bulk refractive indices to calculate the temperature increase of MSPs in the polar summer mesopause. Additionally, we confirmed that the absorption efficiency increases with increasing iron content of potential MSP materials (Asmus et al., 2014).

We find that the impact of solar radiation on polar mesospheric ice particle formation is lower than previously assumed. Critical temperatures for ice growth activation at 69°N decrease at most by 4 K for typical MSP particle sizes. However, for

assessing the significance of solar heating of MSPs on PMC properties, the whole life cycle of mesospheric ice particles has to be considered. Therefore, we propose that the updated ice activation model (Appendix A and B) is used in future model studies with and without solar irradiation for various potential MSP materials, in order to evaluate if absorption of solar irradiation alters properties of polar mesospheric clouds.

**Data availability**

All data are available on request from the corresponding author.

**Appendix A: Critical temperatures under the influence of solar radiation**

At the conditions of the polar summer mesopause, ice particle formation proceeds via deposition of amorphous solid water (ASW) on MSPs (Duft et al., 2018; Nachbar et al., 2018c). Ice growth is activated if the saturation $S(T_p)$ is larger than the critical saturation $S_{cr}(T_p)$. In the following, we present a method for calculating the temperature at which this condition is

fulfilled, the so-called critical temperature.

For a given water vapor mixing ratio $MR$ and atmospheric pressure $p_{atm}$, the saturation is determined by

$$S(T_p) = \frac{MR \cdot p_{atm}}{p_{s,a}(T_p)} \cdot \sqrt{\frac{T_{env}}{T_p}}, \tag{A1}$$

with the saturation vapor pressure of ASW described by (Nachbar et al., 2018c):

$$p_{s,a} = p_{s,h} \cdot exp\left(\frac{2312\,[Jmol^{-1}] - 1.6\,[Jmol^{-1}K^{-1}] \cdot T}{RT}\right) \tag{A2}$$

$p_{s,h}$ represents the saturation vapor pressure of hexagonal ice (Murphy and Koop, 2005). The critical saturation $S_{cr}$ needed to activate ice growth depends on the Kelvin effect calculated at the particle radius $r_{wet}$ (including the amount of adsorbed H$_2$O molecules) and considering the collision radius of a water molecule ($r_{H2O} = 0.15$ nm (Bickes et al., 1975)) (Duft et al., 2018):

$$S_{cr}(T_p) = \left(\frac{r_{wet}}{r_{wet} + r_{H2O}}\right)^2 \cdot exp\left(\frac{2v\sigma}{RT_p r_{wet}}\right) \tag{A3}$$

To determine the critical temperature at which ice growth is activated, the environmental temperature $T_{env}$ is decreased until $S(T_p) \geq S_{cr}(T_p)$. A reasonable starting point for $T_{env}$ is the temperature at which the saturation over a flat surface is 1 (solve for $p_{s,a}(T_{env}) = MR \cdot p_{atm}$). The coupled calculation of $T_p$ and $r_{wet}$ is described in Appendix B and has to be repeated every time $T_{env}$ is decreased. The environmental temperature fulfilling $S(T_p) = S_{cr}(T_p)$ is the critical temperature needed to activate ice growth and $\Delta T = T_p - T_{env}$ is the increase of the particle temperature at conditions of ice particle

formation.




### Appendix B: Equilibrium particle temperature and wet particle radius

Substituting Eq. (6) and Eq. (7) in Eq. (5) and solving for the increase of the particle temperature $\Delta T$ considering the dependency of the solar spectrum and of $Q_{abs}$ on $\lambda$ yields:

$$\Delta \mathrm{T} = T_p - T_{env} = \left(A_{geo} \cdot \int_0^\infty I(\lambda) \cdot Q_{abs}(\lambda, r_{dry}) d\lambda + P_{env}^a - P_{rad}^e\right) \Big/ \left(A_{col} \alpha p v_{th} \frac{(\gamma+1)}{8T(\gamma-1)}\right) \tag{B1}$$

Here, $I(\lambda)$, $P_{env}^a$, and $P_{rad}^e$ were calculated as presented in Asmus et al. (2014). The thermal accommodation coefficient $\alpha$ which is typically used in literature to describe the heating of MSPs or NLC particles is 0.5 (e.g. Asmus et al., 2014; Espy and Jutt, 2002). This value seems to originate from the work of Grams and Fiocco (1977), and was chosen due to a lack of relevant measurements determining $\alpha$ at realistic mesopause conditions. More recently, measurements of $\alpha$ for $N_2$ on water droplets and for air on fused silica have become available which show that $\alpha$ is close to unity (Fung and Tang, 1988; Ganta

et al., 2011). We therefore used $\alpha = 1$. The collision surface area of a particle $A_{col} = 4\pi(r_{wet} + r_{N2})^2$ is described by the collision radius of a nitrogen molecule $r_{N2} = 0.19$ nm (Hirschfelder et al., 1966) and the wet particle radius $r_{wet}$ (Eq. (3)). The wet particle radius can be calculated with the mass of adsorbed water in equilibrium $m_{ads}$ which can be obtained by rearranging Eq. (1):

$$m_{ads} = A_p \cdot m_{H2O} \cdot \frac{n_{H2O} \cdot v_{th}}{4f} \cdot \exp\left(\frac{E_{des}^0}{RT_p} - \frac{2\sigma v}{RT_p r_{dry}}\right) \tag{B2}$$

Note that $m_{ads}$ and $P_{rad}^e$ depend on the particle temperature. Consequently, $\Delta T$ (Eq. (B1)) has to be solved numerically. In our approach we alternatingly calculate $T_p$ and $m_{ads}$ until the relative change of $\Delta T$ is less than 0.01 %).

### Author contribution

MN, DD, HW, KK and TL designed the experiments. MN, TA, and KK carried out the experiments. MN performed the data analysis. HW performed the calculations of the critical temperature in the mesopause. MN prepared the manuscript with

contributions from all co-authors. DD, TM, JP, MR and TL supervised the project.

### Competing interests

The authors declare that they have no conflict of interest.

### Acknowledgements

The authors thank the German Federal Ministry of Education and Research (BMBF, grant number 05K13VH3 and

05K16VHB) and the German Research Foundation (DFG, grant number LE 834/4-1) for financial support of this work. We acknowledge support by the Deutsche Forschungsgemeinschaft and Open Access Publishing Fund of Karlsruhe Institute of Technology. TA is supported by a research studentship from the UK Natural Environment Research Council's SPHERES doctoral training programme.




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
