# Peer review of "The impact of solar radiation on polar mesospheric ice particle formation"

_Atmospheric Chemistry and Physics, 2018_

## Referee Comment (RC1) · Anonymous Referee #1 · 1 Dec 2018

General comments:

This is an excellent study on the impact of solar heating of meteoric smoke particles on the formation of NLC/PMC particles. The work is based on a combination of lab work and simulation results and is an important contribution to improve our understanding of the formation of ice particles near the polar summer mesopause. The manuscript is very well written, the results are substantial and appear to be robust. I did not find much to criticize and in my opinion the paper can be published more or less as is. I have a few minor comments that I ask the authors to consider.

Specific comments:

- Is anything known about the shape of MSP particles in the atmosphere and/or the

MSP analogues in your experiments? If yes, this should be mentioned. I suggest stating in any case that the particles are most likely not spherical.

- Page 1, line 17: I suggest replacing "cold temperatures" by "low temperatures", similarly on line 2 of the following page: "coldest temperatures" -> "lowest temperatures"

- Page 4, line 5: I would explain "Da", despite the fact that many readers will know what it stands for.

- Page 6, line 4: Please explain "ad-layer"

- Page 7, 17: "show a linear trend with particle radius". This is not the most precise way to put it. It may be misinterpreted as a linear relationship between absorption efficiencies and radius, which is not the case.

- Page 8, line 17: add space between degree sign and "(21"

- Same line: only a small point, but shouldn't the SZA be about 45.5 deg at 69 N and for summer solstice?

- Page 11, lines 4 and 5: It would be good to state, which solar irradiance spectrum was used? Why was I(lamda) "calculated" ?
* * *

---

## Referee Comment (RC2) · Anonymous Referee #3 · 31 Jan 2019

This is well written and conclusive study on the effect of solar heating on ice particle formation in the mesopause region. The findings are clearly presented and seem to be sound, although I suggest to include more background information especially when introducing equation (1) to explain the water vapour equilibrium. Also, I suggest to include a little more discussion of the uncertainty in the assumed thermal accommodation coefficient, which is decreased here to 0.5 in comparison to previous studies, where a factor of 1 was used. Does this lead to the reduced solar heating or is the uptake of water molecules (see discussion on p. 9, l. 17-22) the driving factor?

In addition I have a few minor more specific comments:

p.2. l. 3. (e.g. Luebken , 1999 . . .)

[Figure]

p. 1. L. 12.: trends of temperature caused . . .

p.3. l. 16 -27.: is the water measured in the chamber or is it calculated assuming equilibrium conditions. If measured, how are the measurements done?

P5. Top. This section is rather hard to read. I suggest to include an introduction to explain the background and the basic assumptions which went into the formulation of eq. (1).

p.5. l. 7: I'm not sure what is meant by the molar volume of a $H_2O$ molecule. Is it molar or per molecule?

p.5. l. 8: please explain what ASW means.

p. 7. L. 12: I suggest writing "the imaginary part k of the refractive index" for more clarity.

p.8. l. 25: Is this represented by the blue line in Fig 5? Please specify.

Fig. 4: I think that legend might help to improve the readability of the Figure.

---

## Author Comment (AC1) · 4 Mar 2019

The comment was uploaded in the form of a supplement:
https://www.atmos-chem-phys-discuss.net/acp-2018-1032/acp-2018-1032-AC1-supplement.pdf
* * *

---

## Author Response (AR1)

**Response to comments of the Anonymous Referees 1 and 3 on the manuscript entitled "The impact of solar radiation on polar mesospheric ice particle formation"**

We thank the reviewers for the very encouraging review and the thoughtful comments which we address individually below. Page and line numbers in the "Changes made" sections refer to the annotated version of the revised manuscript.

**Comments of Referee 1:**

**General comment:** *This is an excellent study on the impact of solar heating of meteoric smoke particles on the formation of NLC/PMC particles. The work is based on a combination of lab work and simulation results and is an important contribution to improve our understanding of the formation of ice particles near the polar summer mesopause. The manuscript is very well written, the results are substantial and appear to be robust. I did not find much to criticize and in my opinion the paper can be published more or less as is. I have a few minor comments that I ask the authors to consider.*

**Comment 1:** *Is anything known about the shape of MSP particles in the atmosphere and/or the MSP analogues in your experiments? If yes, this should be mentioned. I suggest stating in any case that the particles are most likely not spherical.*
**Response:** Except indications that MSPs are most likely composed of iron-rich materials (Hervig et al., 2017; Rapp et al., 2012) little is known about the shape of these particles in the atmosphere.
However, the shape of metal oxide nanoparticles produced in similar particle sources as used in our experiment has been shown to be compact and spherical (Nachbar et al., 2018a). We added the information about the sphericity of the particles to the manuscript.
**Changes made:**
- Page 3, line 7: We produce **spherical,** singly charged…

**Comment 2:** *Page 1, line 17: I suggest replacing "cold temperatures" by "low temperatures", similarly on line 2 of the following page: "coldest temperatures" -> "lowest temperatures"*
**Response:** We conducted the suggested changes.
**Changes made:**
- Page 1, line 17: The **low** temperatures…
- Page 2, line 2: The **lowest** temperatures…

**Comment 3:** *Page 4, line 5: I would explain "Da", despite the fact that many readers will know what it stands for.*

**Response:** We added the information to the manuscript.

**Changes made:**
- Page 4, line 8: …for the combinations of particle mass ( $2 \cdot 10^4$ Da $- 50 \cdot 10^4$ Da; **1 Da** $\triangleq$ **1 atomic mass unit** $= \mathbf{1.6605 \cdot 10^{-27} kg}$),…

**Comment 4:** *Page 6, line 4: Please explain "ad-layer"*

**Response:** With ad-layer we are referring to the layer of adsorbed water molecules in equilibrium as introduced in the previous section. To be clearer we rephrased the term "ad-layer" to "layer of adsorbed water molecules".

**Changes made:**
- Page 6, line 15: We assume the **layer of adsorbed water molecules** to be entirely transparent…

**Comment 5:** *Page 7, 17: "show a linear trend with particle radius". This is not the most precise way to put it. It may be misinterpreted as a linear relationship between absorption efficiencies and radius, which is not the case.*

**Response:** The absorption efficiency is defined as the absorption cross section divided by the geometric cross section. There is a linear relationship between radius and absorption efficiencies in Rayleigh regime (as investigated in this study). See discussion below Eq. (9) (page 7, line 9-11) or Eq. (3) in (Rapp et al., 2010) (doi:10.1029/2009JD012725).

**No changes made.**

**Comment 6:** *Page 8, line 17: add space between degree sign and "(21". only a small point, but shouldn't the SZA be about 45.5 deg at 69 N and for summer solstice?*

**Response:** We added the space between the degree sign and (. Yes, the SZA was a typo, it is 45.6° and not 46.5°.

**Changes made:**
- Page 9, line 8: For the incoming solar irradiation we use the maximum solar zenith angle of **45.6°** (21[st] June, noon)…

**Comment 7:** *Page 11, lines 4 and 5: It would be good to state, which solar irradiance spectrum was used? Why was I(lambda) "calculated" ?*

**Response:** We assumed black body radiation with a temperature of 5780 K to calculate $I(\lambda)$. The wavelength dependency is needed since the absorption efficiency is wavelength dependent. Thus we have to integrate over $I(\lambda) \cdot Q_{abs}(\lambda)d\lambda$.

**Changes made:**
- Page 11, line 22: Here, $I(\lambda)$ **(black body radiation assuming T=5780 K)**, $P_{env}^a$, and $P_{rad}^e$ were calculated as presented in Asmus et al. (2014).

**Comments of Referee 3:**

**Comment 1:** *This is well written and conclusive study on the effect of solar heating on ice particle formation in the mesopause region. The findings are clearly presented and seem to be sound, although I suggest to include more background information especially when introducing equation (1) to explain the water vapour equilibrium.*

**Response:** We rewrote the paragraph introducing Eq. (1) to improve readability and include more background information.

**Changes made:**
- Page 5, line 4-27: Paragraph rewritten.

**Comment 2:** *Also, I suggest to include a little more discussion of the uncertainty in the assumed thermal accommodation coefficient, which is decreased here to 0.5 in comparison to previous studies, where a factor of 1 was used. Does this lead to the reduced solar heating or is the uptake of water molecules (see discussion on p. 9, l. 17-22) the driving factor?*

**Response:** In the manuscript, we use two different values for the thermal accommodation coefficients $\alpha$:

(1) In the analysis of our experimental data we use the thermal accommodation coefficient of He (the carrier gas in our experiments) which was determined experimentally to $0.525 \pm 0.125$ (Fung and Tang, 1988; Ganta et al., 2011) (see page 6, l. 23-25). The error of the thermal accommodation coefficient on the determined absorption efficiencies is considered in the error analysis of our data as discussed in the manuscript (see page 7, l. 2-4).

(2) In Section 4.2 on the impact of solar radiation on ice particle formation in the mesopause we use the thermal accommodation coefficient of air. Here, we assume $\alpha_{air}=1$. In previous model publications a lower value of $\alpha_{air}=0.5$ was used (e.g. Asmus et al., 2014; Espy and Jutt, 2002). The choice of $\alpha_{air}=0.5$ in these studies appears not to be based on actual experimental data but seems to originate from the work of Grams and Fiocco (1977) who simply estimated $\alpha_{air}$ due to a lack of relevant measurements determining $\alpha$ at realistic mesopause conditions. Some years later two independent experimental studies showed that $\alpha_{air}$ is close to unity as detailed in the manuscript in Appendix B, page 11, lines 23-27. A discussion of the consequences of choosing $\alpha_{air}=1$ instead of 0.5 on the solar heating of the particles is presented in the manuscript on page 10, l. 13-16.

Nevertheless, after re-reading the corresponding paragraphs, we feel that the usage of α may appear ambiguous in the manuscript. We therefore specified the usage of α throughout the manuscript.

**Changes made:**
- Page 6, line 25-27: **Note that when applying the equilibrium temperature model to the summer mesopause in Section 4.2 we use the thermal accommodation coefficient of air $\alpha_{air}=1$ (Fung and Tang, 1988; Ganta et al., 2011).**

**In addition I have a few minor more specific comments:**

**Comment 3:** *p.2. l. 3. (e.g. Luebken , 1999 …)*

**Response:** We added "e.g." to the citation list.

**Changes made:**
- Page 2, line 3: …(**e.g.** Lübken, 1999…)

**Comment 4:** *p. 2. L. 12.: trends of temperature caused …*
**Response:** In this statement we refer to long term trends of temperature and $H_2O$ concentration. We added this information.
**Changes made:**
- Page 2, line 12: for long-term trends **of temperature and $H_2O$ concentration** caused by…

**Comment 5:** *p.3. l. 16 -27.: is the water measured in the chamber or is it calculated assuming equilibrium conditions. If measured, how are the measurements done?*
**Response:** The water vapor concentration in MICE originates from sublimation of water ice covered and temperature controlled surfaces. The saturation vapor pressure of the ice phase on the surfaces was calibrated in one of our previous studies (Nachbar et al., 2018b). We added this information.
**Changes made:**
- Page 3, line 22-23: In MICE, the particles also interact with a **well calibrated (Nachbar et al., 2018b)** concentration of gas-phase $H_2O$ molecules, **which is maintained by temperature-controlled sublimation of water vapor from ice covered surfaces** (Duft et al., 2015).

**Comment 6:** p.5. *Top. This section is rather hard to read. I suggest to include an introduction to explain the background and the basic assumptions which went into the formulation of eq. (1).*
**Response:** See our response to comment 1.
**Changes made:**
- Page 5, line 4-27: Paragraph rewritten.

**Comment 7:** *p.5. l. 7: I'm not sure what is meant by the molar volume of a H2O molecule. Is it molar or per molecule?*
**Changes made:**
- Page 5, line 19: … is the volume of **one mole of** $H_2O$ molecules in ASW…

**Comment 8:** *p.5. l. 8: please explain what ASW means.*
**Response:** We now introduce the meaning of ASW (amorphous solid water) in the rewritten paragraph on page 5.
**No changes made.**

**Comment 9:** *p. 7. L. 12: I suggest writing "the imaginary part k of the refractive index" for more clarity.*
**Response:** We adapted the passage according to the suggestion.
**Changes made:**
- Page 8, line 5: …that the **values of the imaginary part** k **of the refractive index** of Querry (1985) and…

**Comment 10:** *p.8. l. 25: Is this represented by the blue line in Fig 5? Please specify.*
**Response:** We added more explanations to this paragraph in order to express our conclusion more clearly.
**Changes made:**
- Page 10, line 1-6: The horizontal dotted line at T=130 K indicates the measured mean temperatures at 87 km and 69°N during June and July (Lübken, 1999). **This line intersects with the calculations for non-absorbing particles at a dry particle radius of about 1 nm, which means that at the mean temperature of 130 K particles larger than $r_{dry}$=1 nm will activate ice growth.** The solid red curve in Fig. 5a shows results obtained when including solar heating of the particles ($T_p > T_{env}$) yielding lower $T_{cr,env}$ values. **With solar heating no particles will activate at the mean particle temperature of 130 K.** However, typical temperature variations **in the summer mesopause** are on the order of 10 K (Rapp et al., 2002) **which leads us to** conclude that the atmospheric temperature of the summer mesopause frequently falls below the critical temperature of non-absorbing **as well as of absorbing** MSPs **for** particle radii above 0.5 nm.

**Comment 11:** *Fig. 4: I think that legend might help to improve the readability of the Figure.*
**Response:** We included this suggestion in the revised manuscript.
**Changes made:**
- Page 8, Figure 4: We added a legend to Figure 4.

[revised manuscript text omitted]